# Paradoxical Changes: EMMPRIN Tissue and Plasma Levels in Marfan Syndrome-Related Thoracic Aortic Aneurysms

**DOI:** 10.3390/jcm13061548

**Published:** 2024-03-08

**Authors:** Kyle C. Alexander, Carlton W. Anderson, Chris B. Agala, Panagiotis Tasoudis, Elizabeth N. Collins, Yiwen Ding, John W. Blackwell, Danielle E. Willcox, Behzad S. Farivar, Melina R. Kibbe, John S. Ikonomidis, Adam W. Akerman

**Affiliations:** 1Department of Surgery, Division of Cardiothoracic Surgery, University of North Carolina at Chapel Hill, Chapel Hill, NC 27599, USA; kyle_alexander@med.unc.edu (K.C.A.); chris_agala@med.unc.edu (C.B.A.); yiwen_ding@med.unc.edu (Y.D.); john_blackwell@med.unc.edu (J.W.B.); john_ikonomidis@med.unc.edu (J.S.I.); 2Advanced Analytics Core, Center for Gastrointestinal Biology and Disease, University of North Carolina at Chapel Hill, Chapel Hill, NC 27514, USA; 3Department of Biomedical Engineering, University of Virginia, Charlottesville, VA 22908, USAerv2vm@uvahealth.org (M.R.K.); 4Department of Surgery, University of Virginia, Charlottesville, VA 22908, USA

**Keywords:** EMMPRIN, MT1-MMP, Marfan syndrome, thoracic aortic aneurysm

## Abstract

**Background:** Thoracic aortic aneurysms (TAAs) associated with Marfan syndrome (MFS) are unique in that extracellular matrix metalloproteinase inducer (EMMPRIN) levels do not behave the way they do in other cardiovascular pathologies. EMMPRIN is shed into the circulation through the secretion of extracellular vesicles. This has been demonstrated to be dependent upon the Membrane Type-1 MMP (MT1-MMP). We investigated this relationship in MFS TAA tissue and plasma to discern why unique profiles may exist. **Methods**: Protein targets were measured in aortic tissue and plasma from MFS patients with TAAs and were compared to healthy controls. The abundance and location of MT1-MMP was modified in aortic fibroblasts and secreted EMMPRIN was measured in conditioned culture media. **Results**: EMMPRIN levels were elevated in MFS TAA tissue but reduced in plasma, compared to the controls. Tissue EMMPRIN elevation did not induce MMP-3, MMP-8, or TIMP-1 expression, while MT1-MMP and TIMP-2 were elevated. MMP-2 and MMP-9 were reduced in TAA tissue but increased in plasma. In aortic fibroblasts, EMMPRIN secretion required the internalization of MT1-MMP. **Conclusions**: In MFS, impaired EMMPRIN secretion likely contributes to higher tissue levels, influenced by MT1-MMP cellular localization. Low EMMPRIN levels, in conjunction with other MMP analytes, distinguished MFS TAAs from controls, suggesting diagnostic potential.

## 1. Introduction

Thoracic aortic aneurysms (TAAs) occur because of the abnormal remodeling of the aortic extracellular matrix (ECM). Vascular remodeling is a process in which a family of proteolytic enzymes, the matrix metalloproteinases (MMPs), actively degrade the vessel wall and subsequently release sequestered enzymes, growth factors, and cytokines [1,2]. This progressive breakdown of normally long-lasting matrix molecules emphasizes the importance of understanding the role of MMPs and their endogenous inhibitors, the tissue inhibitors of MMPs (TIMPs). The differential expression of MMPs and TIMPs occurs in TAAs, contributing to pathology. Extracellular MMP inducer (EMMPRIN), also known as CD147, is a widely expressed glycoprotein that stimulates the production of various MMPs; EMMPRIN exists in cell membrane-spanning and secreted forms. MMPs drive pathological remodeling; EMMPRIN stimulates the production of MMPs, suggesting a direct role in pathology.

EMMPRIN is released into circulation through the secretion of extracellular vesicles and this process has been demonstrated to be dependent upon hydrolysis by the Membrane Type-1 MMP (MT1-MMP), also known as MMP-14 [3,4]. Both MT1-MMP and EMMPRIN are elevated in TAA tissue [5,6,7]. In other diseases, EMMPRIN abundance is often increased in both the pathological tissue and circulation [8,9,10,11,12,13]. However, a recent study concerning Marfan syndrome (MFS) patients with aortic ectasia found the opposite, i.e., circulating plasma levels of EMMPRIN were low [8]. While EMMPRIN’s specific role in pathology remains poorly understood, low levels of circulating EMMPRIN are pathognomonic and can predict ectasia in MFS patients. The tissue levels and underlying mechanisms, however, warrant further investigation.

MFS is a connective tissue disorder with an incidence of 1:5000. It is caused by mutations that result in malformed microfibrils, which are key structural components involved in the packaging and secretion of proteins and nucleic acids in extracellular vesicles [9,10,11]. This study investigates EMMPRIN in tissues and plasma to test the hypothesis that EMMPRIN levels are elevated in MFS TAA tissue and reduced in circulation.

## 2. Materials and Methods—See Appendix A for Details

Study population: MFS ascending TAA tissue and plasma were previously collected by the GenTAC registry and transferred to our laboratory by the NHLBI Biologic Specimen and Data Repository Information Coordinating Center. Healthy human plasma was purchased from Boca Biolistics. Secondary analysis is not considered as human subject research. See Appendix A for patient information.

Multiplex Suspension Array (MSA): MSA measured MMPs and TIMPs in human plasma and aortic tissues (FCSTM07 and LKTM003, R&D Systems, Minneapolis, MN, USA). For aortic tissue, 20 µg of total protein was analyzed. Plasma dilutions were as follows: MMP-2 and MMP-9, 1:100; all other MMPS, 1:10; TIMPs, 1:20.

Immunoblotting analysis: Aortic tissues were homogenized [12]. A total of 10 µg of total protein was analyzed for MT1-MMP abundance (1:2000; ab38971, Abcam, Cambridge, UK). Gels were normalized using two methods. First, following MT1-MMP analysis, nitrocellulose membranes were washed with a harsh stripping buffer. The membrane was incubated in antisera specific for GAPDH (1:2500; Cat#31460, Thermo Fisher Scientific, Waltham, MA, USA). Second, normalization was performed using total protein densitometry (ultraviolet light-activated trihalo) between the 20 and 120 kDa molecular weights. The protein levels of GAPDH were compared to the total protein densitometry. The coefficient of variation (ratio of standard deviation to the mean) was performed on each normalization method. Densitometric quantitation of total protein, measured after gel transfer to nitrocellulose, exhibited a lower coefficient of variation compared to GAPDH. Therefore, total protein densitometry was selected for the normalization of immunoblot data.

Cell Culture: Primary aortic fibroblast cell lines (*n* = 3) were established using human aortic biopsies from non-syndromic, non-aneurysmal, ascending thoracic aortae using a previously described outgrowth technique [12]. The fibroblasts were maintained in complete fibroblast-specific growth media (Fibroblast Growth Media 2; C-23020, PromoCell, Heidelburg, Germany) with added 10% Premium Grade Fetal Bovine Serum (Avantor, Seradigm 97068-085), and gentamicin (0.5 mg/mL; 15710-064, Gibco, Waltham, MA, USA) at 37 °C in 5% CO_2_. Culture passages 2 to 10 were used in experiments.

In Vitro Transfection: Aortic fibroblasts were transfected for 18 h using jetPRIME transfection reagent (114-15, Polypolus-transfection Illkirch-Graffenstaden, France) according to the DNA transfection protocol with either 10 µg MT1-MMP over expression vector (MC219253, OriGene, Rockville, MD, USA) or the transfection reagent alone. Following the 18 h incubation, the culture media was replaced with fresh, serum-free media for 48 hours. EMMPRIN levels were quantified in the conditioned culture media using the human EMMPRIN ELISA (ab219631, Abcam, Cambridge, UK).

Alteration of MT1-MMP Localization: MT1-MMP localization was altered using methods previously described [5]. Treatment with phorbol 12-myristate 13-acetate (PMA) activates both the classical and novel forms of protein kinase C (PKC), internalizing MT1-MMP. Röttlerin, a PKC-δ-specific inhibitor, can lock MT1-MMP on the cell surface. Aortic fibroblasts were treated with PMA (100 nM), Röttlerin (3 µM), or vehicle control (DMSO, 1:1000) and the conditioned, serum-free media was collected after 24 h. EMMPRIN levels were quantified in the conditioned culture media using the human EMMPRIN ELISA (ab219631, Abcam).

Data Analysis: Statistical analyses were performed using SYSTAT (SigmaPlot—Statistical Analysis version 15). All data were assessed for normality using the Shapiro–Wilk test and the corresponding appropriate statistical tests and sample sizes with *p* values are listed in each respective figure legend and summarized in Appendix A. In all figures, raw data (dots) are displayed next to the group mean, median, and standard error of the mean. Effect size (ES) calculations were performed using Cohen’s d or Hedges’ g test. For each ES value, an effect level was assigned using the following scheme: <0.1: Trivial; 0.1–0.3: Small; 0.3–0.5: Medium; >0.5: Large. Multivariate logistic regression modeling with forward selection was used to identify the best predictors of TAA presence. Once the best predictors were identified, we used logistic regression to generate Receiver Operating Characteristic (ROC) curves, area under the curve, and the associated 95% confidence intervals and *p*-values. For all comparisons, a *p* value of <0.05 was considered significant.

## 3. Results—See Appendix A for Details

We present a secondary analysis conducted on a subset of clinical specimens originally collected within the framework of the Genetically Triggered Thoracic Aortic Aneurysms and Cardiovascular Conditions (GenTAC) study. While we did not design the original study, we have endeavored to analyze the data with novel, quantitative methods in order to maximize its scientific utility.

### 3.1. EMMPRIN Levels in MFS TAA Tissue and Plasma

EMMPRIN levels were measured in aortic tissue and plasma from MFS patients with TAAs and were compared to healthy controls. EMMPRIN abundance was increased in MFS TAA tissue (2.14 ± 0.324 fold, *n* = 10) when compared to controls (1.00 ± 0.18, *n* = 6) (*p* = 0.023; ES: 1.25, ESL: Large) (Figure 1A). Circulating EMMPRIN levels were decreased in MFS TAA plasma (0.640 ± 0.0325 fold, *n* = 46) when compared to controls (1.00 ± 0.0715, *n* = 115) (*p* = 0.041; ES: 0.543, ESL: Large) (Figure 1B). To investigate sex as a biological variable, plasma EMMPRIN levels were compared between males and females. No difference was detected between males and females in healthy controls (males: 3479.41 ± 376.63, *n* = 57, age = 41 ± 16 years; females: 3152.20 ± 301.34, *n* = 58, age = 42 ± 18 years). Interestingly, in MFS TAAs, lower levels of EMMPRIN were detected in the plasma of females versus males (females: 1888.01 ± 145.64, *n* = 19, age = 38 ± 3 years; males: 2326.34 ± 155.31, *n* = 27, age = 37 ± 3 years; *p* = 0.039; ES: 0.603 ESL: Large). No difference in circulating EMMPRIN levels was detected in other TAA subtypes: Bicuspid Aortic Valve (BAV) (2896.32 ± 355.506, *n* = 57), Familial TAA (FTAA) (2432.08 ± 316.258, *n* = 29), Turner syndrome (TS) (4645.46 ± 1804.28, *n* = 28), and Ehlers–Danlos (ED) (3142.164 ± 603.188, *n* = 8), when compared to controls (3314.37 ± 239.102, *n* = 115) (*p* = 0.194) (Figure 1C).

### 3.2. MT1-MMP Is Elevated in MFS TAA Tissue

Because MT1-MMP is involved with the release of EMMPRIN from cells, it was quantified in MFS TAA tissue and compared to controls. Following exposure to anti-sera specific for MT1-MMP, the optical intensity of the 64 kDa active form of MT1-MMP was quantified and normalized to total protein levels. Representative image: Figure 2A. MT1-MMP protein abundance was increased 1.847 ± 0.295 fold in MFS TAA tissue when compared to controls (*n* = 7, *p* = 0.014; ES: 2.36, ESL: Large) (Figure 2B).

### 3.3. Gelatinase Abundance in MFS TAA Tissue and Plasma

MT1-MMP is involved in the release of cell surface-associated gelatinases (MMP-2 directly and MMP-9 indirectly) [13,14]. Gelatinase abundance was measured in MFS TAA tissue and plasma and was compared to controls. Active MMP-2 abundance trended lower in MFS TAA tissue (0.322 ± 0.072 fold, *n* = 10) when compared to controls (1.000 ± 0.339, *n* = 5) (*p* = 0.116; ES: 1.38, ESL: Large); however, statistical significance was not achieved (Figure 3A). Circulating MMP-2 levels were significantly increased in MFS TAA plasma (2.527 ± 0.164 fold, *n* = 44) when compared to controls (1.00 ± 0.0375, *n* = 110) (*p* = <0.001; ES: 2.28, ESL: Large) (Figure 3B). MMP-9 abundance trended lower in MFS TAA tissues (0.594 ± 0.710 fold, *n* = 10) when compared to controls (1.00 ± 0.342, *n* = 5) (*p* = 0.327; ES: 0.53, ESL: Large); however, statistical significance was not achieved (Figure 3C). Circulating MMP-9 levels were significantly increased in MFS TAA plasma (2.002 ± 0.316 fold, *n* = 36) when compared to controls (1.00 ± 0.0823, *n* = 50) (*p* = 0.006; ES: 0.76, ESL: Large) (Figure 3D). These findings suggest that MT1-MMP is predominantly located on the cell surface in MFS TAA tissue, because that is the location where MMP-2 and MMP-9 are hydrolyzed and released into circulation [15].

### 3.4. Stromelysin and Collagenase Levels in MFS TAA Tissue and Plasma

Elevated circulating EMMPRIN levels stimulate the cellular production of stromelysin and collagenases [16]. MMP-3 and MMP-8 are prototypical members of the stromelysin and collagenase subtypes, respectively. Protein abundance was measured in MFS TAA tissue and plasma and was compared to healthy controls. No difference in MMP-3 levels was detected in MFS TAA tissues (1.98 ± 0.916 fold, *n* = 10) when compared to controls (1.00 ± 0.239, *n* = 6) (*p* = 0.588; ES: 0.395, ESL: Medium) (Figure 4A). Circulating MMP-3 levels were decreased in MFS TAA plasma (0.716 ± 0.0767 fold, *n* = 46) when compared to controls (1.00 ± 0.0739, *n* = 114) (*p* = 0.038; ES: 0.414 ESL: Medium) (Figure 4B). MMP-8 was decreased in MFS TAA tissues (0.428 ± 0.187 fold, *n* = 10) when compared to controls (1.00 ± 0.150, *n* = 4) (*p* = 0.048; ES: 1.00, ESL: Large) (Figure 4C), while no differences in circulating MMP-8 levels were detected in MFS TAA plasma (0.981 ± 0.096 fold, *n* = 46) when compared to controls (1.00 ± 0.0828, *n* = 113) (*p* = 0.894; ES: 0.029, ESL: Trivial) (Figure 4D). These findings are consistent with decreased circulating levels of EMMPRIN in MFS TAA plasma.

### 3.5. TIMP Levels in MFS TAA Tissue and Plasma

An increased MMP activity may reflect an increase in the absolute amount of these enzymes or a relative loss of MMP inhibitory control by naturally occurring TIMPs (TIMP-1 being most common in the aorta) [17]. TIMP-1 was quantified in MFS TAA tissue and plasma and compared to controls. No difference in TIMP-1 abundance was detected in MFS TAA tissues (0.849 ± 0.095 fold, *n* = 10) when compared to controls (1.00 ± 0.147, *n* = 6) (*p* = 0.382; ES: 0.44, ESL: Medium) (Figure 5A), while circulating TIMP-1 levels were reduced in MFS TAA plasma (0.735 ± 0.043 fold, *n* = 30) when compared to controls (1.00 ± 0.0423, *n* = 35) (*p* < 0.001; ES: 1.09, ESL: Large) (Figure 5B). These findings suggest upregulated proteolytic activity of the MMPs in MFS TAA tissue opposed to a relative loss of MMP inhibitory control.

TIMP-2 plays a different role that includes the cell surface formation of a trimolecular complex involving binding to MT1-MMP and the incorporation of pro-MMP-2 [13]. The subsequent release of active MMP-2 was quantified in MFS TAA tissue and plasma and was compared to controls. TIMP-2 abundance was increased in MFS TAA tissues (1.658 ± 0.164 fold, *n* = 8) when compared to controls (1.00 ± 0.166, *n* = 5) (*p* = 0.022; ES: 1.42, ESL: Large) (Figure 5C), while circulating levels were decreased in MFS TAA plasma (0.474 ± 0.0214 fold, *n* = 32) when compared to controls (1.00 ± 0.063, *n* = 40) (*p* ≤ 0.001; ES: 1.70, ESL: Large) (Figure 5D). Because we observed an increase in the plasma levels of active MMP-2, the elevation of TIMP-2 in the tissue further suggests an increased cell surface localization of MT1-MMP in MFS.

### 3.6. Receiver Operator Characteristic (ROC) Curve Analysis of Plasma Analytes

ROC curve analysis of plasma analyte concentrations determined the diagnostic viability and predictability. The area under the curve (AUC) for EMMPRIN levels was 0.603 (95% CI: 0.5189–0.6875, *n* = 161, *p* = 0.0164) (Figure 6A). A forward stepwise multivariable approach determined the best predictive panel, the order being TIMP-2, MMP-2, and MMP-1, respectively (AUC 0.996) (95% CI: 0.9879–1.0000, *n* = 48, *p* < 0.0001) (Figure 6B). Excluding TIMP-1 and TIMP-2 provides access to a larger sample size. ROC curve analysis using EMMPRIN and the MMPs in the order of EMMPRIN, MMP-2, MMP-3, and MMP-8 yields an AUC of 0.962 (95% CI: 0.9277–0.9954, *n* = 154, *p* < 0.0001) (Figure 6C).

### 3.7. Effects of MT1-MMP Abundance and Localization on Secreted EMMPRIN Levels In Vitro

Because the isolation of living fibroblasts from cryopreserved tissues in the GenTAC repository is not possible, we demonstrated a potential mechanistic relationship between MT1-MMP and the cellular secretion of EMMPRIN in healthy, primary human thoracic aortic fibroblasts. Then, 48 h following transfection with an MT1-MMP over expression vector, EMMPRIN levels were measured in conditioned culture media. EMMPRIN levels were decreased in the culture media from MT1-MMP over expressing fibroblasts (80.600 ± 2.566, *n* = 3) when compared to controls (108.267 ± 2.315, *n* = 3) (*p* = 0.001; ES: 5.22 ESL: Large) (Figure 7A). Findings demonstrate that an increased abundance of MT1-MMP alone does not increase EMMPRIN release/secretion from aortic fibroblasts.

The localization of MT1-MMP regulates function by controlling the access to substrates (i.e., EMMPRIN). MT1-MMP was internalized using PMA or held on the cell surface using Röttlerin. EMMPRIN levels were measured in conditioned culture media and compared to controls. No statistical difference was detected, but secreted EMMPRIN levels were higher in PMA-treated culture media (266.600 ± 3.745, *n* = 3). After the internalization of MT1-MMP, EMMPRIN levels were significantly decreased in conditioned culture media from Röttlerin-treated fibroblasts (188.933 ± 4.947, *n* = 3) when compared to controls (236.047 ± 17.805, *n* = 3) (*p* = 0.044; ES: 1.37 and 2.08, ESL: Large and Large). These findings demonstrate that the internalization of MT1-MMP from the plasma membrane is required for the cellular secretion of EMMPRIN from aortic fibroblasts.

## 4. Discussion

EMMPRIN levels were elevated in MFS TAA tissue but reduced in plasma, compared to controls. Low circulating EMMPRIN levels, in conjunction with other MMP analytes, distinguished MFS TAAs from controls, suggesting a diagnostic potential. In MFS, impaired EMMPRIN secretion likely contributes to higher tissue levels influenced by MT1-MMP cellular localization. This appears to be demonstrative of the idea that cellular location is a key factor in aneurysm development. Indeed, most drugs are designed and operate on the foundational premise that molecules exist in distinct cellular locations. The fact that molecules themselves appear to change location because of pathology is not well understood. Targeting cellular location as a therapeutic—specifically, altering where molecules go and how long they are allowed to stay there—is both exceptionally novel and exciting. As we have shown from experiments and the literature, moving molecules like MT1-MMP and EMMPRIN to different cellular locations regulates their access to substrates. Controlling access via location also controls (changes) function (ability to function in a certain capacity). From 2006 to 2016, the NHLBI and the National Institute of Arthritis and Musculoskeletal and Skin Diseases jointly funded the GenTAC registry to gather data and specimens from patients confirmed or suspected to have a genetic condition that raised their risk of developing thoracic aortic aneurysms. After the completion of this study, the registry made specimens available for secondary analyses and subsequently provided this laboratory with all available data and specimens for the explicit purpose of maximizing their scientific value.

This study’s central paradox is as follows: in MFS TAA tissue, MT1-MMP and EMMPRIN levels do not behave as expected; that is, they both increase in MFS TAA tissue, meaning EMMPRIN should also be elevated in circulation. In MFS, however, the opposite is true. To investigate further, we changed the location of MT1-MMP in aortic fibroblasts. Aortic fibroblasts are a key constituent of the aortic wall and maintain vessel structure and function [18]. We found that MT1-MMP must be internalized from the plasma membrane for the cell to secrete EMMPRIN in aortic fibroblasts.

EMMPRIN levels go up in the circulation of patients with acute myocardial infarction, coronary artery disease, cancer, and multiple and systemic sclerosis [17,19,20,21,22,23]. We detected lower circulating EMMPRIN levels in MFS TAA patients compared to healthy controls, and we observed lower circulating EMMPRIN levels in female MFS TAA patients, indicating that sex is a biological variable. A prior study in MFS patients with aortic ectasia also found that circulating EMMPRIN levels were reduced [8]. When comparing circulating EMMPRIN levels to healthy controls, lower values proved predictive of TAAs in patients with MFS (AUC: 0.603). This AUC is similar to that found in the study examining ectasia (AUC: 0.763). It asserted that circulating EMMPRIN levels, when combined with established tools, such as transthoracic echocardiography, computed tomography, and magnetic resonance imaging, can effectively monitor the progression of the aortic diameter and that the additional biochemical information can further discern risk [24,25].

EMMPRIN is abundantly expressed in the cells and pathological tissues associated with TAAs [7]. Our study found that EMMPRIN levels were elevated in MFS TAA tissue, suggesting that, in MFS, the secretion of EMMPRIN is blocked, causing higher EMMPRIN abundance in the cells and tissue. This, however, does not appear to induce MMP expression. Specifically, high tissue abundance coincided with either no change, or even a reduction in MMP-3, MMP-8, and TIMP-1.

An increased abundance of MT1-MMP is important in pathological remodeling. Once activated, MT1-MMP degrades a wide portfolio of substrates [26,27,28,29]. Because MT1-MMP is involved with the release of EMMPRIN from cells [4], it seems paradoxical that this elevation coincides with a reduction in circulating levels in MFS TAA plasma.

When on the cell surface, MT1-MMP activates and releases other MMPs into circulation. A prototypical example is the formation of a trimolecular complex involving the binding of MT1-MMP and TIMP-2, and the incorporation of pro-MMP-2, by which MT1-MMP directly liberates active MMP-2, and indirectly MMP-9, into circulation. We observed an increase in MT1-MMP and TIMP-2 abundance in MFS TAA tissue. Our results suggest that, in MFS TAA tissue, MT1-MMP is predominantly located on the cell surface, as evidenced by the reduced tissue levels of MMP-2 and MMP-9, alongside increased plasma levels.

MT1-MMP is preferentially internalized from the cell membrane in endosomal vesicles following phosphorylation at amino acid threonine-567 by PKC [5]. When internalized, induction results in a cascade of activity [30,31,32]. This places MT1-MMP at the center of several mechanisms capable of regulating pathological remodeling. Elucidating the endogenous mechanisms that regulate MT1-MMP’s location will provide foundational insights that contribute to therapeutic development.

In the early stages of non-heritable TAAs, MT1-MMP accumulates on the surface of aortic fibroblasts [5]. While on the plasma membrane, MT1-MMP hydrolyses ECM substrates. Later, in non-heritable TAA development, MT1-MMP is preferentially internalized, suggesting a multifaceted role in disease progression. The location of MT1-MMP seems critical to functionality. In our examination of MFS, results suggest that MT1-MMP is predominantly located on the cell surface. MT1-MMP and EMMPRN need caveolae to move inside the cell; caveolae formation depends upon the proper assembly of microfibrils [33,34,35,36]. Microfibrils are an assemblage of fibrillin-1. In MFS, there is less fibrilin-1 available and/or it is malformed. Fibrilin-1, then, may be a prerequisite for caveolae, which then affects the function and location of MT1-MMP and EMMPRN because they do not internalize into the cell as they would in homeostatic circumstances.

To recapitulate the increased abundance of MT1-MMP in vitro, we over expressed MT1-MMP in healthy aortic fibroblasts using a pCMV6-AC expression vector. The results mimicked those found in MFS TAA tissue and plasma. An increased MT1-MMP resulted in low levels of EMMPRIN being released into the culture media. Interestingly, this outcome was the same as another study that used opposite methodology, an siRNA knocked-down MT1-MMP [4]. This is perplexing. We hypothesize that this can be resolved by considering the temporal location of MT1-MMP following transfection. Studies involving healthy aortic fibroblasts show that endogenous levels of MT1-MMP are comparatively low [5]. Furthermore, MT1-MMP is predominantly located on the cell surface 48 hours after over expression (these studies transfected healthy aortic fibroblasts with green florescent labeled MT1-MMP that was expressed via the same pCMV6-AC vector) [5]. When we forced the internalization of MT1-MMP, or held it on the cell surface, we observed inverse effects on the release of EMMPRIN. PMA treatment increased EMMPRIN release while Röttlerin decreased it. While treatment with PMA did not reach statistical significance, others performing similar studies have [3,4]. Taken together, these findings demonstrate that the internalization of MT1-MMP from the plasma membrane is required for the cellular secretion of EMMPRIN from aortic fibroblasts.

This study is not without limitations. Because of the length of time and methods used to cryopreserve the MFS TAA tissues, isolation of live fibroblasts is not possible. Therefore, aortic fibroblast cultures from non-syndromic, non-TAA aortic tissues were selected for the analysis of MT1-MMP’s relationship with EMMPRIN secretion. An increased age may alter MMP and TIMP levels [37]. Studies identifying the proteolytic profiles of differently aged people should provide further insight. Comparisons were made between MFS patients with TAA and also non-TAA and non-MFS tissues. We recognize that the use of non-TAA MFS tissue would be ideal for comparison, but these tissues are rare and difficult to acquire. Additionally, further studies will be necessary for comparison between patients affected by non-MFS aortic aneurysms. Because ROC curve analyses are confined to the lowest analyte sample size (in our case the TIMP values), statistical power can be skewed, making interpretation less reliable. To increase the statistical power and provide a robust analysis specific to MFS syndrome, we excluded the TIMP values from the final model. Nevertheless, AUC values proved highly sensitive and specific.

## 5. Conclusions

The implications of these findings are as follows: First, it is imperative that the mechanisms by which cellular location changes—that is, their function via access to cellular substrates—in aneurysm disease be studied and understood. Second, circulating EMMPRIN levels are inherently and abundantly valuable as a diagnostic tool, one which, in MFS patients, may be capable of monitoring for aneurysm presence or have the potential to determine aneurysm severity, both of which will better inform decisions regarding surgical resection. Third, the different heritable forms of TAAs appear to cause discrete pathological consequences, suggesting that TAA disease is far more complex than we believe it to be, and that no single treatment is ever likely to remedy its myriad manifestations. Fourth, therapies, therefore, must be designed such that mechanistic intervention into the localization pathway is paramount. The one we posit in this discussion exemplifies the idea that regulating the cellular location of molecules is a mechanistic intervention that modifies the pathway underlying aortic tissue remodeling.

## Figures and Tables

**Figure 1 jcm-13-01548-f001:**
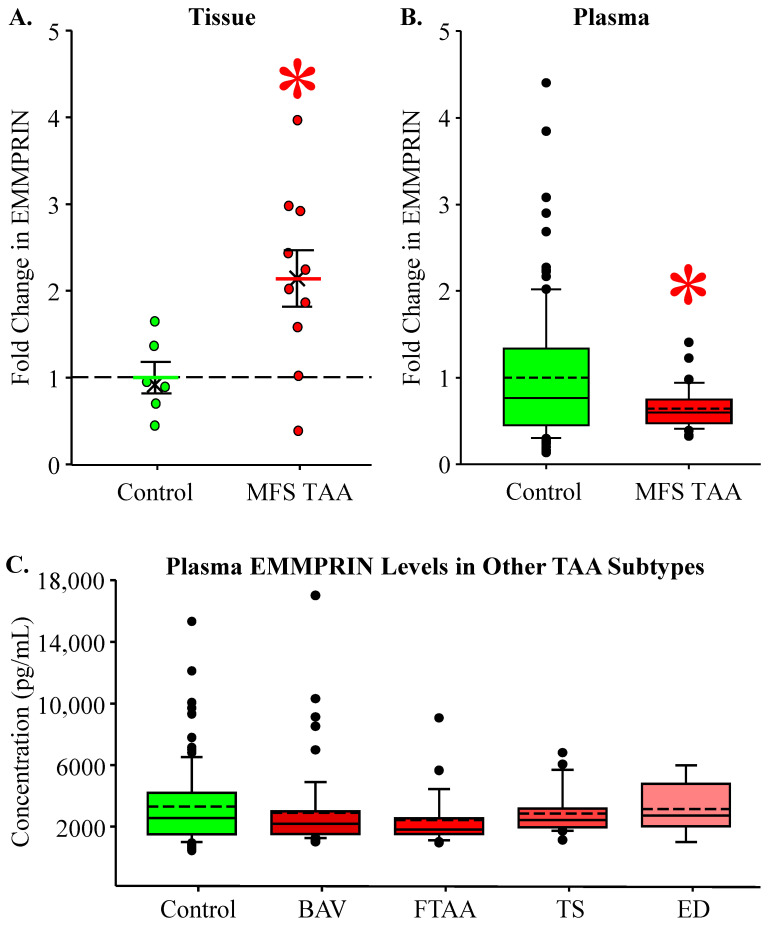
EMMPRIN levels in MFS TAA tissue and plasma. (**A**) Fold change in EMMPRIN abundance in aortic tissues. Control is green; MFS TAA is red. The mean is depicted as a solid bar ± the standard error of the mean. The median is depicted as an X. All data points are shown. * *p* < 0.05 vs. control (Student’s *t*-test). (**B**) Fold change in EMMPRIN abundance in plasma. Data are displayed as box and whisker plots where the box signifies the 25th–75th interquartile range, and both the median (solid line) and the mean (dashed line) values are shown within the box. The whiskers are defined as the data points with the highest or lowest value below quartile 3 + 1.5 times the interquartile range. Outliers are shown and defined as values that fall above or below the interquartile range. * *p* < 0.05 vs. control (Student’s *t*-test). (**C**) Plasma EMMPRIN concentration (pg/mL) in patients without TAAs (control, green bar) and in other TAA subtypes. Bicuspid Aortic Valve (BAV), Familial TAA (FTAA), Turner syndrome (TS), and Ehlers–Danlos (ED). The dashed line is the mean of the control group. The colored bars signify mean concentrations ± standard error of the means. No statistical differences were detected.

**Figure 2 jcm-13-01548-f002:**
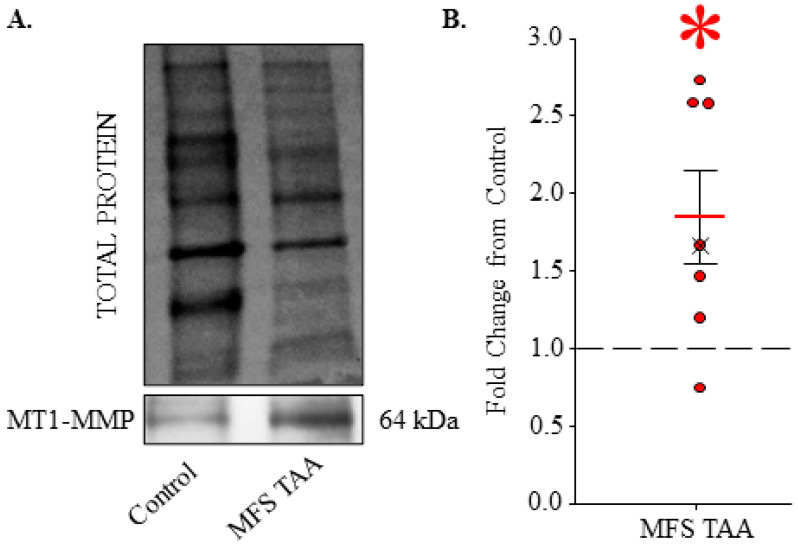
MT1-MMP is elevated in MFS TAA Tissue. (**A**) Representative total protein and immunoblot for the full-length, active form of MT1-MMP (64 kDa) abundance in control and MFS TAA tissue. (**B**) Fold change in MT1-MMP abundance in MFS TAA tissue compared to control tissues (dashed line). The mean is depicted as a solid red bar ± the standard error of the mean. The median is depicted as an X. All data points are shown. * *p* < 0.05 vs. control (Student’s *t*-test).

**Figure 3 jcm-13-01548-f003:**
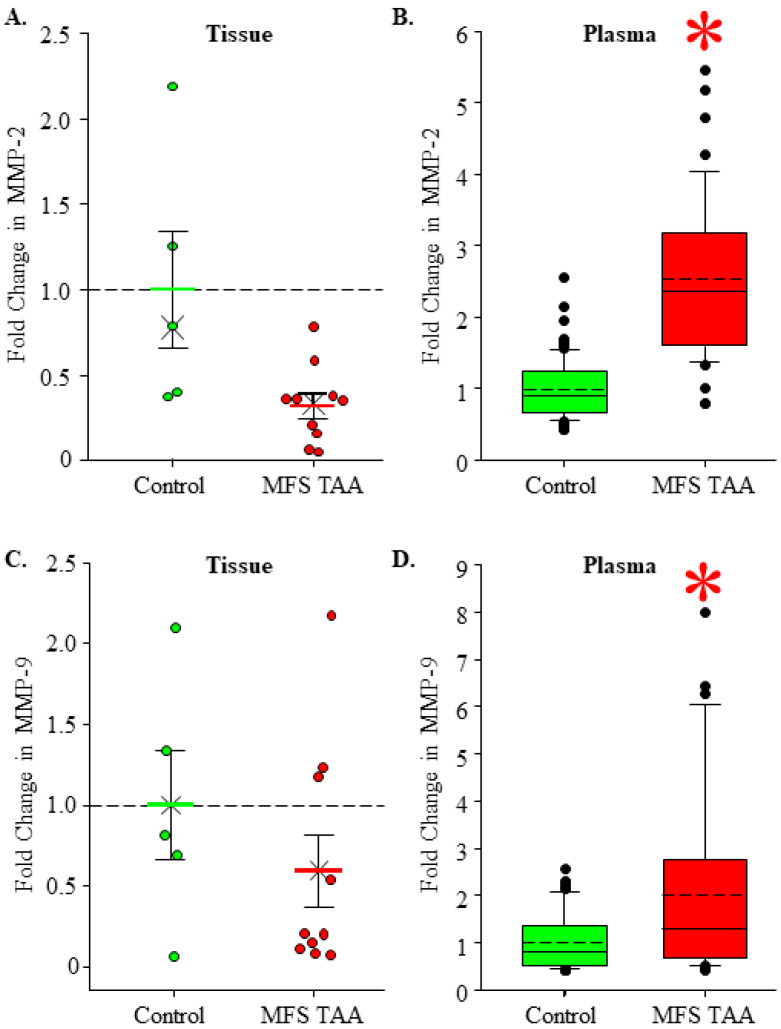
Gelatinase abundance in MFS TAA tissue and plasma. (**A**) Fold change in MMP-2 abundance in aortic tissues. Control is green; MFS TAA is red. The mean is depicted as a solid bar ± the standard error of the mean. The median is depicted as an X. All data points are shown. (**B**) Fold change in MMP-2 abundance in plasma. Data are displayed as box and whisker plots where the box signifies the 25th–75th interquartile range, and both the median (solid line) and the mean (dashed line) values are shown within the box. The whiskers are defined as the data points with the highest or lowest value below quartile 3 + 1.5 times the interquartile range. Outliers are shown and defined as values that fall above or below the interquartile range. * *p* < 0.05 vs. control (Student’s *t*-test). (**C**) Fold change in MMP-9 abundance in aortic tissues. Control is green; MFS TAA is red. The mean is depicted as a solid bar ± the standard error of the mean. The median is depicted as an X. All data points are shown. No statistical differences were detected. (**D**) Fold change in MMP-9 abundance in plasma. Data are displayed as box and whisker plots where the box signifies the 25th–75th interquartile range, and both the median (solid line) and the mean (dashed line) values are shown within the box. The whiskers are defined as the data points with the highest or lowest value below quartile 3 + 1.5 times the interquartile range. Outliers are shown and defined as values that fall above or below the interquartile range. * *p* < 0.05 vs. control (Student’s *t*-test).

**Figure 4 jcm-13-01548-f004:**
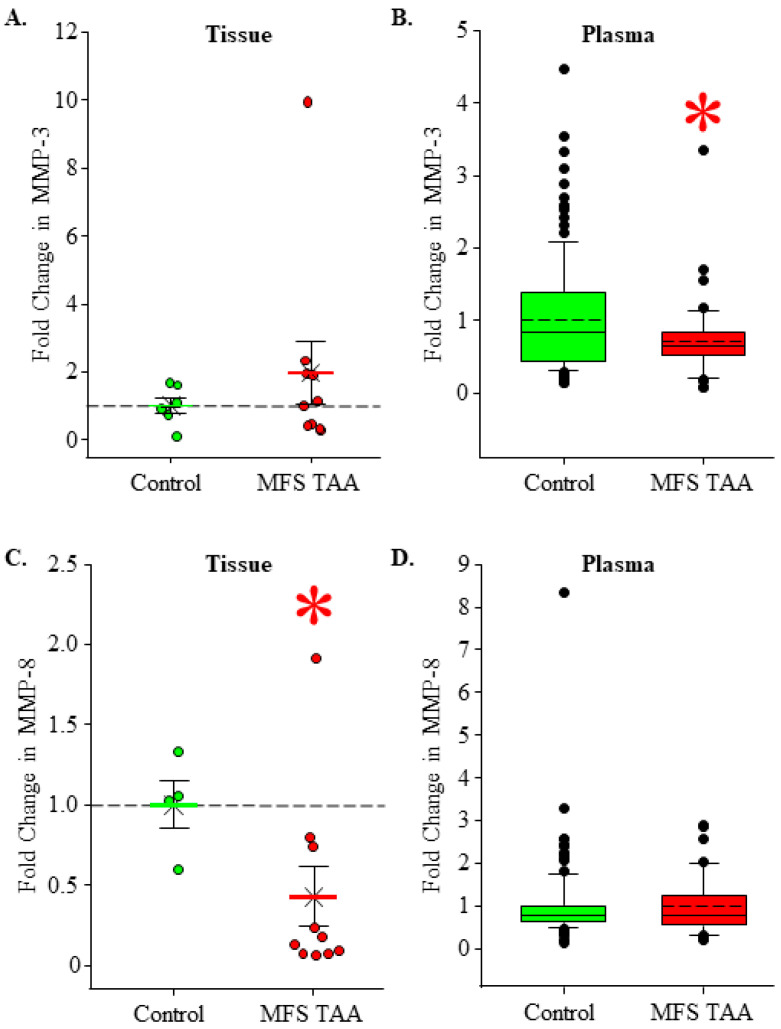
Stromelysin and collagenase levels in MFS TAA tissue and plasma. (**A**) Fold change in MMP-3 abundance in aortic tissues. Control is green; MFS TAA is red. The mean is depicted as a solid bar ± the standard error of the mean. The median is depicted as an X. All data points are shown. No statistical differences were detected. (**B**) Fold change in MMP-3 abundance in plasma. Data are displayed as box and whisker plots where the box signifies the 25th–75th interquartile range, and both the median (solid line) and the mean (dashed line) values are shown within the box. The whiskers are defined as the data points with the highest or lowest value below quartile 3 + 1.5 times the interquartile range. Outliers are shown and defined as values that fall above or below the interquartile range. * *p* < 0.05 vs. control (Student’s *t*-test). (**C**) Fold change in MMP-8 abundance in aortic tissues. Control is green; MFS TAA is red. The mean is depicted as a solid bar ± the standard error of the mean. The median is depicted as an X. All data points are shown. * *p* < 0.05 vs. control (Student’s *t*-test). (**D**) Fold change in MMP-8 abundance in plasma. Data are displayed as box and whisker plots where the box signifies the 25th–75th interquartile range, and both the median (solid line) and the mean (dashed line) values are shown within the box. The whiskers are defined as the data points with the highest or lowest value below quartile 3 + 1.5 times the interquartile range. Outliers are shown and defined as values that fall above or below the interquartile range. No statistical differences were detected.

**Figure 5 jcm-13-01548-f005:**
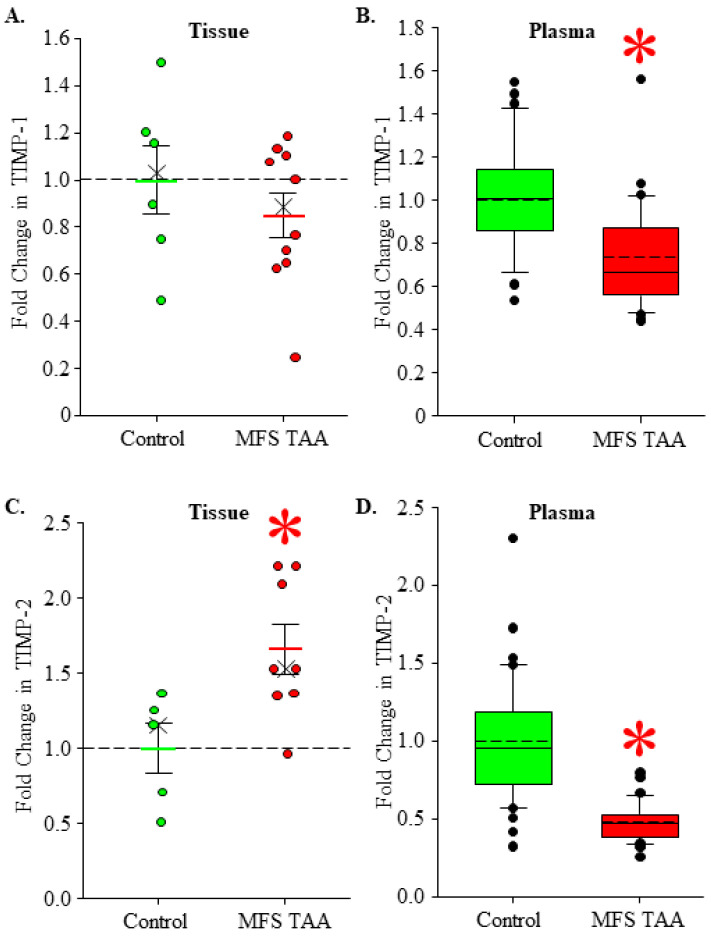
TIMP levels in MFS TAA tissue and plasma. (**A**) Fold change in TIMP-1 abundance in aortic tissues. Control is green; MFS TAA is red. The mean is depicted as a solid bar ± the standard error of the mean. The median is depicted as an X. All data points are shown. No statistical differences were detected. (**B**) Fold change in MMP-3 abundance in plasma. Data are displayed as box and whisker plots where the box signifies the 25th–75th interquartile range, and both the median (solid line) and the mean (dashed line) values are shown within the box. The whiskers are defined as the data points with the highest or lowest value below quartile 3 + 1.5 times the interquartile range. Outliers are shown and defined as values that fall above or below the interquartile range. * *p* < 0.05 vs. control (Student’s *t*-test). (**C**) Fold change in MMP-8 abundance in aortic tissues. Control is green; MFS TAA is red. The mean is depicted as a solid bar ± the standard error of the mean. The median is depicted as an X. All data points are shown. * *p* < 0.05 vs. control (Student’s *t*-test). (**D**) Fold change in MMP-8 abundance in plasma. Data are displayed as box and whisker plots where the box signifies the 25th–75th interquartile range, and both the median (solid line) and the mean (dashed line) values are shown within the box. The whiskers are defined as the data points with the highest or lowest value below quartile 3 + 1.5 times the interquartile range. Outliers are shown and defined as values that fall above or below the interquartile range. * *p* < 0.05 vs. control (Student’s *t*-test).

**Figure 6 jcm-13-01548-f006:**
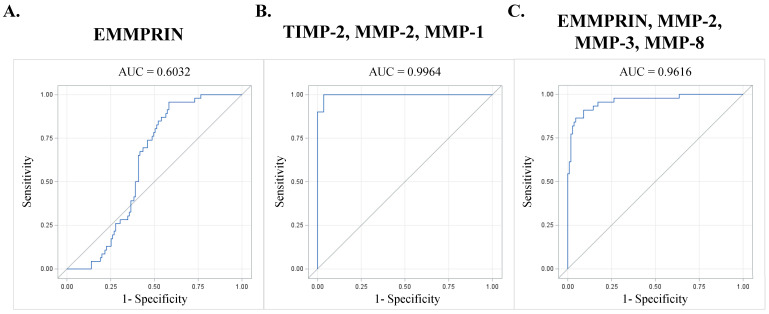
ROC curve analysis for the forward stepwise variable selection. (**A**) Receiver Operating Characteristic (ROC) curve plot for EMMPRIN, depicting the relationship between sensitivity and 1-specificity. (**B**) ROC curve plot for TIMP-2, MMP-2, and MMP-1, demonstrating their diagnostic potential. (**C**) ROC curve plot for EMMPRIN, MMP-2, MMP-3, and MMP-8, indicating their combined predictive value. The area under the curve (AUC) is displayed above each plot, representing the diagnostic accuracy. This figure provides insights into the diagnostic predictability of different analytes and variable selection approaches, aiding in the assessment of disease detection in clinical settings. Note: ROC curves visually represent the performance of diagnostic tests, with AUC values reflecting the overall accuracy.

**Figure 7 jcm-13-01548-f007:**
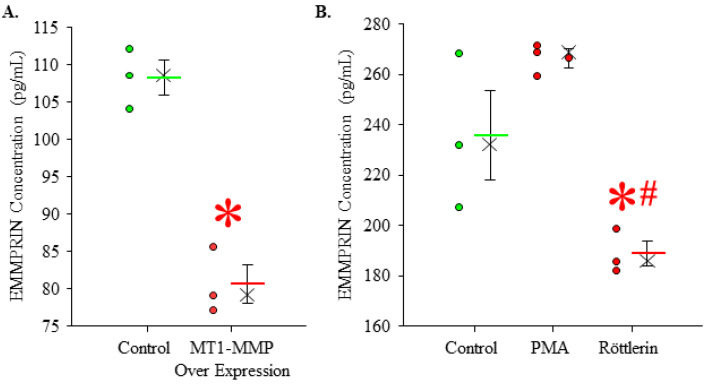
Effects of MT1-MMP abundance and localization on secreted EMMPRIN levels in conditioned culture media. (**A**) Concentration of EMMPRIN in conditioned culture media from aortic fibroblasts transfected with an MT1-MMP over expression vector compared to the control. The mean is depicted as a solid bar ± the standard error of the mean. The median is depicted as an X. All data points are shown. * *p* < 0.05 vs. control (Student’s *t*-test). (**B**) Concentration of EMMPRIN in conditioned culture media from aortic fibroblasts treated with PMA or Röttlerin compared to the control. The mean is depicted as a solid bar ± the standard error of the mean. The median is depicted as an X. All data points are shown. * *p* < 0.05 vs. control (ANOVA, post hoc test: Holm–Sidak); # *p* < 0.05 vs. PMA (ANOVA, post hoc test: Holm–Sidak).

## Data Availability

All relevant data are within the manuscript and its Appendix A.

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
