# Peer review of "Paradoxical Changes: EMMPRIN Tissue and Plasma Levels in Marfan Syndrome-Related Thoracic Aortic Aneurysms"

_jcm, 2024, doi:10.3390/jcm13061548_

Round 1

Reviewer 1 Report

Comments and Suggestions for Authors

The article is very interesting and the evidence could, in the long run, lead to really interesting results. 

Authors reported, in the limitation of the study, the main structural defects of the article. 

- Figure 1: add an explanation of the statistical tables and the abbreviations inserted

- The results section is verbose and difficult to fully understand for non-genetic experts. Considering the difficulties of the topic, the presence of all the abbreviations makes reading even more difficult. You should simplify and shorten it.

- limitation of the study: I would add to the section that: further studies will be necessary to carrying out a comparison with patients affected by non-Marfan aortic aneurysm. 

Comments on the Quality of English Language

Minor editing of English language required

Author Response

Thank you for your careful consideration. We have addressed all comments below and have improved the manuscript.

-Figure 1: add an explanation of the statistical tables and the abbreviations inserted –

Abbreviations used in the table are defined in the figure legend.

The figure legend is as follows: “A. Fold change in EMMPRIN abundance in aortic tissues. Control is green; MFS TAA is red. The mean is depicted as a solid bar ± the standard error of the mean. The median is depicted as an X. All data points are shown. * p < 0.05 vs Control (Student’s T-test). B. Fold change in EMMPRIN abundance in plasma. Data is displayed as box and whisker plots where the box signifies the 25th – 75th interquartile range, and both the median (solid line) and the mean (dashed line) values are shown within the box. The whiskers are defined as the data points with the highest or lowest value below quartile 3 + 1.5 times the interquartile range. Outliers are shown and defined as values that fall above or below the interquartile range. * p < 0.05 vs Control (Student’s T-test). C. Plasma EMMPRIN concentration (pg/mL) in patients without TAA (control, green bar) and in other TAA subtypes. Bicuspid Aortic Valve (BAV), Familial TAA (FTAA), Turner syndrome (TS), and Ehlers-Danlos (ED). The dashed line is the mean of the control group. The colored bars signify mean concentrations ± standard error of the means. No statistical differences were detected.”

We have now included a summary of all statistics for all figures in an excel file entitled “Summary of Statistics for All Figures.” Summary statistics are explained in tab 1 of the excel file labeled “Stat.Summary”.

-The results section is verbose and difficult to fully understand for non-genetic experts. Considering the difficulties of the topic, the presence of all the abbreviations makes reading even more difficult. You should simplify and shorten it.

We have ensured that any needlessly verbose usage has been removed from the results section.

-limitation of the study: I would add to the section that: further studies will be necessary to carrying out a comparison with patients affected by non-Marfan aortic aneurysm.

Comparisons were made between MFS patients with TAA and non-TAA, non-MFS tissues. We recognize that use of non-TAA MFS tissue would be ideal for comparison, but these tissues are rare and difficult to acquire. Additionally, further studies will be necessary for comparison between patients affected by non-MFS aortic aneurysm.

Reviewer 2 Report

Comments and Suggestions for Authors

I am grateful to the editors for the opportunity to review the manuscript by Alexander et al. “Paradoxical changes: EMMPRIN tissue and plasma levels in Marfan syndrome-related thoracic aortic aneurysm.” In this article, the authors examined the hypothesis that that EMMPRIN levels are elevated in MFS TAA tissue and reduced in the circulation. Indeed, the authors showed that compared to controls, EMMPRIN levels are increased in MFS TAA tissue and decreased in plasma. These data confirmed the previous results of Rurali et al that MFS patients are characterized by lower sEMMPRIN levels than HC. Notably, plasma sEMMPRIN levels are strongly associated with thoracic AE (see ref. 8 in the article). The article also obtained new scientific results, that in MFS, EMMPRIN secretion from cells through vesicles is likely blocked, leading to higher EMMPRIN abundance in MFS TAA tissue, which likely involves the cellular location of MT1-MMP. Also, when combined with other MMP analytes, low levels of EMMPRIN were highly sensitive and specific for distinguishing MFS TAA from healthy controls. The authors were so inspired by the results obtained (“The manifold implications of these findings are immense” - line 420) that they did not carefully format the text of the manuscript.

My questions and comments:

1. The first paragraph in the introduction (lines 33-37) is inappropriate. The introduction should begin with information justifying the relevance of the study, and not about the study design and methods of obtaining data. This is more appropriate in section 2. Materials and Methods.

2. Since tissue samples were obtained from patients, and the article was submitted to a clinical journal, information about patients and the control group should be presented in the main text of the manuscript, and not as a Supplemental file. In addition, the information provided in this file is insufficient; it is required to present it in the form of a table with information about statistical differences between the main group and the control.

3. The Discussion section also needs to be improved. It should begin with the main results obtained by the authors.

4. Section 5. Conclusions requires revision. I believe that the phrase “The manifold implications of these findings are immense” is inappropriate in this scientific article; its language should be more academic. This assessment can be given by reviewers and commentators, but not by the authors themselves. The rest of the text in this section cannot be a conclusion (conclusions) from the materials of the article. It is most appropriate for inclusion in the Discussion section as an illustration of the scientific and clinical significance of this study. Section 5. Conclusions should include the data obtained in this study (for example, in the same style as they are presented in the Abstract.

Comments on the Quality of English Language

No comments

Author Response

Response to review:

We would like to thank reviewer 2 for their thoughtful examination of our manuscript. Their insightful critiques have helped us improve it significantly.

The first paragraph in the introduction (lines 33-37) is inappropriate. The introduction should begin with information justifying the relevance of the study, and not about the study design and methods of obtaining data. This is more appropriate in section 2. Materials and Methods.

As this article is a unique case, it is understandable that it may not conform perfectly to rubicized, sectional constraints and findings and their contextualization were presented in ways that made the manuscript compelling and readable. However, we have rewritten lines 35-37 to be more appropriate. They now read:

“We present a secondary analysis conducted on a subset of clinical specimens originally collected within the framework of the Genetically Triggered Thoracic Aortic Aneurysm and Cardiovascular Conditions (GenTAC) study. While we did not design the original study, we have endeavored to analyze the data with novel, quantitative methods in order to maximize its scientific utility.”

Since tissue samples were obtained from patients, and the article was submitted to a clinical journal, information about patients and the control group should be presented in the main text of the manuscript, and not as a Supplemental file. In addition, the information provided in this file is insufficient; it is required to present it in the form of a table with information about statistical differences between the main group and the control.

­­­ We have now included a detailed summary table of all statistics showing comparisons versus controls. Also, we have chosen to leave this information in the supplement because it is not directly relevant to the main body of the article or to the conclusions it draws. As such, it would obfuscate the article itself, negatively affecting readability. Moreover, the supplement is easily accessible as it is. A limitation of this study, as described in the limitations section of the manuscript, is that it does not constitute a ‘perfect’ like-for-like comparison, and the patient information is essentially supplemental in nature. Therefore, including wholly supplemental information in the main body of the text would be counterproductive in our opinion.

The Discussion section also needs to be improved. It should begin with the main results obtained by the authors.

A concise statement of results obtained by the authors has been added to the beginning of the Discussion section:

      “EMMPRIN levels were elevated in MFS TAA tissue but reduced in plasma compared to controls. Low circulating EMMPRIN levels, in conjunction with other MMP analytes, distinguished MFS TAA from controls, suggesting diagnostic potential. In MFS, impaired EMMPRIN secretion likely contributes to higher tissue levels influenced by MT1-MMP cellular localization. This appears to be demonstrative of the idea that cellular location is a key factor in aneurysm development. Indeed, most drugs are designed and operate on the foundational premise that molecules exist in distinct cellular locations. The fact that molecules themselves appear to change location because of pathology is not well understood. Targeting cellular location as a therapeutic—specifically, altering where molecules go and how long they are allowed to stay there—is both exceptionally novel and exciting. As we’ve shown in experiments and literature, moving molecules like MT1-MMP and EMMPRIN to different cellular locations regulates their access to substrates. Controlling access via location also controls (changes) function (ability to function in a certain capacity).”

Section 5. Conclusions requires revision. I believe that the phrase “The manifold implications of these findings are immense” is inappropriate in this scientific article; its language should be more academic. This assessment can be given by reviewers and commentators, but not by the authors themselves. The rest of the text in this section cannot be a conclusion (conclusions) from the materials of the article. It is most appropriate for inclusion in the Discussion section as an illustration of the scientific and clinical significance of this study. Section 5. Conclusions should include the data obtained in this study (for example, in the same style as they are presented in the Abstract.

The phrase “The manifold implications of these findings are immense” is florid and we appreciate that certain readers may be put off by it. As such, we have revised line 424 to read: “The implications of these findings are:” Also, while we understand and appreciate this reviewer’s primary aim in this comment, we wrote the conclusions this way to best represent the article’s most salient and natural conclusions. We would be amenable to revising them should the journal’s editors recommend we do so, however.

Round 2

Reviewer 1 Report

Comments and Suggestions for Authors

Authors basically fixed all the point

Author Response

We are thankful to this reviewer for their thoughtful input. It has improved the manuscript. 

Reviewer 2 Report

Comments and Suggestions for Authors

The authors revised the text of the manuscript and answered my questions. However, I was not satisfied with the authors' response to my first comment. I believe that an article cannot begin with information about methods for obtaining data. The introduction must justify the relevance of the study. I do not think that the possibility of secondary analysis using new quantitative methods alone justifies the relevance of the study.

Comments on the Quality of English Language

No comments

Author Response

We have removed this section from the introduction.